# Role of microglia in the dissemination of Zika virus from mother to fetal brain

Pei Xu[1], Chao Shan[2], Tiffany J. Dunn[1], Xuping Xie[2], Hongjie Xia[2], Junling Gao[1], Javier Allende Labastida[1], Jing Zou[2], Paula P. Villarreal[3], Caitlin R. Schlagal[1], Yongjia Yu[4], Gracie Vargas[1,3], Shannan L. Rossi[5,6,7,8], Nikolaos Vasilakis[5,7,8,9], Pei-Yong Shi[2], Scott C. Weaver[5,6,8,9]*, Ping Wu[1]*

1 Department of Neuroscience, Cell Biology and Anatomy, University of Texas Medical Branch, Galveston, Texas, United States of America, 2 Department of Biochemistry and Molecular Biology, University of Texas Medical Branch, Galveston, Texas, United States of America, 3 Center for Biomedical Engineering, University of Texas Medical Branch, Galveston, Texas, United States of America, 4 Department of Radiology and Oncology, University of Texas Medical Branch, Galveston, Texas, United States of America, 5 Institute for Human Infections and Immunity, University of Texas Medical Branch, Galveston, Texas, United States of America, 6 Department of Microbiology and Immunology, University of Texas Medical Branch, Galveston, Texas, United States of America, 7 Department of Pathology, University of Texas Medical Branch, Galveston, Texas, United States of America, 8 Center for Biodefense and Emerging Infectious Diseases, University of Texas Medical Branch, Galveston, Texas, United States of America, 9 World Reference Center for Emerging Viruses and Arboviruses, University of Texas Medical Branch, Galveston, Texas, United States of America

* sweaver@utmb.edu (SCW); piwu@utmb.edu (PW)

**Data Availability Statement:** All relevant data are within the manuscript.

**Funding:** This work was supported by NIH grants to P.W. (R21AI129509-01), S.C.W. (AI120942), P-Y.S. (AI142759, AI127744, and AI136126). Other

## Abstract

Global Zika virus (ZIKV) outbreaks and their link to microcephaly have raised major public health concerns. However, the mechanism of maternal-fetal transmission remains largely unknown. In this study, we determined the role of yolk sac (YS) microglial progenitors in a mouse model of ZIKV vertical transmission. We found that embryonic (E) days 6.5-E8.5 were a critical window for ZIKV infection that resulted in fetal demise and microcephaly, and YS microglial progenitors were susceptible to ZIKV infection. Ablation of YS microglial progenitors significantly reduced the viral load in both the YS and the embryonic brain. Taken together, these results support the hypothesis that YS microglial progenitors serve as "Trojan horses," contributing to ZIKV fetal brain dissemination and congenital brain defects.

## Author summary

ZIKV is more likely to cause fetal demise and brain malformations when the mother is infected at an early stage of pregnancy, which is the critical time window when a special type of immune cells called microglia appear in the YS and migrate to the fetal brain. YS-derived microglia are susceptible to ZIKV infection and can act as "Trojan horses" to bring ZIKV from the mother to the fetal brain.

funding sources include UTMB CRO Special Fund (P.W.), the John S. Dunn Foundation (P.W., S.C. W., P-Y.S.), the Kleberg Foundation (P-Y.S.), the Amon G. Carter Foundation (P-Y.S.), the Gilson Longenbaugh Foundation (P-Y.S.), the McLaughlin Fund (P.X.), the Summerfield Robert Foundation (P-Y.S.), CONACYT-COPOCYT (J.A.L.), ConTex (J. A.L.) and Fundacion Marron Cajiga (J.A.L.). The funders had no role in study design, data collection and analysis, decision to publish, or preparation of the manuscript.

**Competing interests:** The authors have declared no competing financial interests.

## Introduction

It is well established that ZIKV infection in pregnant women may lead to severe neurological consequences, such as microcephaly in newborns [1–5]. Clinical and animal studies have shown that miscarriage and brain malformation are more frequent when infection occurs early during pregnancy [1–10]. ZIKV-associated microcephaly is most likely due to the high susceptibility of neural stem/progenitor cells (NS/PCs), which populate and develop the fetal brain, to ZIKV infection [11–14]. Nevertheless, how ZIKV gains access to the fetal brain and why earlier infection has more severe outcomes have not been fully elucidated.

Microglia are the first-line defenders against infections in the brain [15–17]. They originate exclusively from the erythro-myeloid progenitors in the YS and migrate to the embryonic brain during early development [18]. A previous study demonstrated that a flavivirus entry factor Axl renders human fetal brain microglia permissive to ZIKV infection [19]. In a human induced pluripotent stem cell (hiPSC)-derived human macrophage/microglia and NS/PC co-culture system, ZIKV-infected macrophages/microglia are able to transmit the virus to NS/PCs and induce apoptosis [20]. However, it is unknown whether YS microglia progenitor cells are susceptible to ZIKV infection. Two research groups have hypothesized that microglia may act as a "Trojan horse" by disseminating Toxoplasma gondii (one of the TORCH agents causing human microcephaly) and bovine viral diarrhea virus (another flavivirus causing congenital infection and fetal demise in cattle) into the brain during invasion [21, 22]. In this study, we determined the role of YS-derived microglial progenitors in a mouse model of vertical ZIKV transmission.

## Methods

### Mice

This study was conducted at the University of Texas Medical Branch, and was reviewed and approved by the Institutional Animal Care and Use Committee of University of Texas Medical Branch (approval 1008041C). All procedures followed the National Institutes of Health Guide for the Care and Use of Laboratory Animals. IFNAR1$^{-/-}$ female mice (8- to 11-week-old) were crossed to C57BL/6J male mice [23], and housed in a specific-pathogen-free facility. Timed matings were set up, and the presence of a vaginal plug in the breeding female was designated as gestational age E0.5. Pregnant dams were infected subcutaneously with $10^4$ PFU ZIKV at E6.5-E8.5, E9.5-E10.5 or E13.5–15.5. Mice were euthanized at E18.5 and dam serum and brain, as well as fetuses, placentas and amniotic fluid, were harvested. For the YS-derived microglia progenitor ablation study, anti-CSF1R mAb (Clone AFS98, BioXCell, West Lebanon, NH, USA) or the rat IgG2a isotype control (Clone 2A3, BioXCell, West Lebanon, NH, USA) was injected intraperitoneally at E6.5 and E7.5 into the pregnant mice with or without subcutaneous inoculation of ZIKV ($10^4$ PFU) at E6.5 or E8.5. Mice were euthanized at E11.5 and dam serum and brain, as well as YS, embryos and placenta were harvested.

### Zika virus

The infectious cDNA clone of ZIKV Puerto Rico strain PRVABC59 (rPRV) use used to rescue challenge virus according to our previous description [24]. The parental PRVABC59 strain (GenBank number KU501215) was obtained from World Reference Center of Emerging Viruses and Arboviruses with five rounds passage on Vero cells. All procedures for handling ZIKV were approved by the Institutional Biosafety Committee.

## Cell culture and infection

YS-derived progenitor cells were isolated from E10.5 pregnant mice as previously described with slight modifications [25]. Briefly, the embryos were removed from the uterus and placed in a sterile dish with ice-cold phosphate-buffered saline (PBS) (Gibco, 21600–05, USA) containing 10% fetal bovine serum (FBS) (HyClone, SH30072.03, USA). The placenta was carefully removed from the embryo-encased YS, then the YS was gently separated from the embryo and placed in ice-cold PBS containing 10% FBS for dissociation. YSs from 5–6 embryos were pooled, passed through 18 gauge followed by 26 gauge needles several times, and then resuspended in PBS containing 0.25% collagenase (Gibco, 17101–015, USA) and 20% FBS. The mixture was incubated at 37˚C for 20 minutes to generate a single-cell suspension. Cells were cultured in Dulbecco's modified Eagle's medium/Ham's F-12 (DMEM/F12) (Gibco, 11330–032, USA), supplemented with 10%FBS, IL-34 (10 ng/mL, R&D Systems, 5195-ML-101, USA), M-CSF(10 ng/mL, R&D Systems, 416-ML-010, USA) and 1% penicillin/ streptomycin (Gibco, 15-140-122, USA); and incubated at 37˚C with 5% $CO_2$ [26, 27]. For ZIKV infections, cells were treated with ZIKV at a multiplicity of infection of 1 Vero PFU/cell for 1 hour.

## Quantitative Reverse Transcription PCR (RT-qPCR)

Total RNA was extracted from tissues or liquid samples using the RNeasy Mini Kit (Qiagen, Hilden, Germany). ZIKV RNA levels were determined by the QuantiTect Probe RT-PCR Kit (Qiagen, Hilden, Germany) according to the manufacturer's manual on the LightCycler 480 System (Roche, Basel, Germany). The real-time PCR primers for ZIKV RNA detection were: ZIKV_1193F: 5′-CCGCTGCCCAACACAAG-3′ and ZIKV_1269R: 5′-CCACTAACGTTCT TTTGCAGACAT-3′ [24]. The probe was 5′-FAM/AGCCTACCT/ZEN/TGACAAGCAAT CAGACACTCAA/3IABkFQ-3' [24]. The one-step RT-qPCR program was 30 minutes at 50˚C for cDNA synthesis, 15 minutes at 95˚C for initial heat activation, and 45 cycles of PCR (94˚C for 15 seconds and 60˚C for 1 minute). ZIKV RNA copies were determined relative to a standard curve produced using serial 10-fold dilutions of *in vitro* transcribed full-length ZIKV RNAs with a known concentration [24].

## Immunofluorescence imaging

Embryos were fixed in 4% paraformaldehyde (Sigma- Aldrich, P6148, USA)-PBS at 4˚C for 1 h and immersed in 10%, 20% and 30% sucrose (Sigma- Aldrich, S7903, USA) until saturation. Tissues were embedded in OCT and fast frozen in a cold bath of methyl butane. Cryosections were cut at 20 μm. The YS was fixed in 4% paraformaldehyde-PBS at room temperature for 1 h and permeabilized in 100% methanol (Fisher Scientific, A412-4, USA) for 30 minutes at −20˚C. Sections of the embryo heads and YSs were blocked in Tris Buffered Saline (TBS) (Sigma- Aldrich, T6664, USA) plus 10% normal goat serum (Jackson Immuno Research, 005-000-121, USA), 0.25% Triton-X-100 (Fisher, BP151-100, USA) and 2% bovine serum albumin (BSA) (Sigma Aldrich, A-4503, USA). Tissues were incubated with primary antibodies overnight at 4˚C, followed by incubation with secondary antibodies in 0.25% Triton-X-100/TBS for 3 h at room temperature. Primary antibodies were included: rabbit antibodies against ZIKV E protein (1:200, Ab00230-23.0, Absolute Antibody, Oxford, UK); rat anti-F4/80 (1:200, ab6640, Abcam) for YS-derived microglia. Secondary antibodies were goat anti-rabbit IgG (1:1000, R37116, Invitrogen) conjugated with Alexa Fluor 488, and goat anti-Rat IgG (1:1000, A-11077, Invitrogen) with Alexa Fluor 568. DAPI (4',6-diamidino-2-phenylindole) was used to stain nuclei at a concentration of 1:5,000. Images were viewed and captured by a Nikon D-Eclipse C1si inverted confocal microscope with the EZ-C1 software v3.50 (Nikon, Japan).

## Plaque assay

Viral titers in the culture medium were determined by standard cytopathic effect-based plaque assay on Vero cells [28]. Briefly, Vero cells ($2 \times 10^5$ per well) were seeded into 24-well plates. After 24 h post-seeding, viral samples were 10-fold serially diluted five times in Dulbecco's modified Eagle's medium (DMEM) (11965–092, Gibco, CA, USA). For each dilution, 100 μl sample was added to one well of the 24-well plate containing 90% confluent Vero cells. The infected cells were incubated at 37˚C in 5% $CO_2$ for 1 h and shacked every 15 mins to ensure even infection. After the incubation, 500 μl of methyl cellulose overlay was added to each well, and the plates were placed into the incubator at 37˚C in 5% $CO_2$. After four days incubation, methyl cellulose overlay was removed, and the plates were fixed with 3.7% formaldehyde at room temperature for 20 mins. Following fixation, the plates were stained with 1% crystal violet for 5 mins. Visible plaques were counted to calculate the viral titers (PFU/mL).

## Statistical analysis

All data were analyzed by GraphPad Prism 6 software and presented as the mean ± SD. Changes of maternal body weight were analyzed by two-way ANOVA with a Tukey post hoc test. Fetal viability data were analyzed with a Chi-square test. Viral RNA data and morphology measurements were analyzed by non-parametric Kruskal Wallis test with Dunn's multiple comparisons or one-way ANOVA with a Tukey post hoc test. A *P* value of $<0.05$ was considered statistically significant.

## Results

### Maternal ZIKV infection on E6.5–8.5 had a higher risk of fetal demise and brain malformation

Since interferon type I receptor-deficient (Ifnar1$^{-/-}$) mice are susceptible to ZIKV infection and disease [23], we initially crossed female Ifnar1$^{-/-}$ mice to Ifnar1$^{-/-}$ or wild-type (WT) males, and subcutaneously infected them with cDNA clone-derived ZIKV (rPRV, an infectious clone of Puerto Rico strain PRVABC59 (24), $10^4$ PFU) on E6.5-E8.5, E9.5-E10.5, or E13.5–15.5 (Fig 1A). These embryonic stages were chosen for their equivalence to the first and second trimesters of pregnancy in human [29]. The miscarriage rate was significantly higher in Ifnar1$^{-/-}$ females crossed to Ifnar1$^{-/-}$ males than in those crossed to WT males (50% versus 23%) (Fig 1B). Thus, the choice of using WT sires allowed us to generate enough embryos to address the critical issue of the role of microglia during ZIKV infection and embryonic brain development. More importantly, the 23% miscarriage rate (7 out of 31 pregnant dams did not have any fetus) of our Ifnar1$^{-/-}$×WT model is closer to that of the nonhuman primates model, which has the miscarriage rate of 26% when infected at early gestation [30].

For assessment of ZIKV-infected fetal brains and clinical presentations, Ifnar1$^{-/-}$ mice crossed with WT males were selected and dams were infected with rPRV ($10^4$ PFU). The dams infected during early pregnancy (E6.5-E8.5) gained less weight than those infected at the later stages, or mock-infected (Fig 1C), probably due to the higher rate of resorption and abnormal embryo development. Inspection of the fetuses on E18.5 revealed that ZIKV infection on E6.5-E8.5 resulted in miscarriage and a significantly higher rate of embryo resorption, compared to those in dams infected at later stages of pregnancy (Fig 1D and 1E). The remaining grossly intact fetuses had much smaller brains than the controls (Fig 1F and 1G). Furthermore, fetal brains from dams infected with ZIKV on E6.5-E8.5 had much higher viral loads detected by RT-qRCR (Fig 2A), whereas both placenta and amniotic fluid had similar high viral loads when infected in all gestation stages (Fig 2B and 2C). Moreover, no significantly higher viral

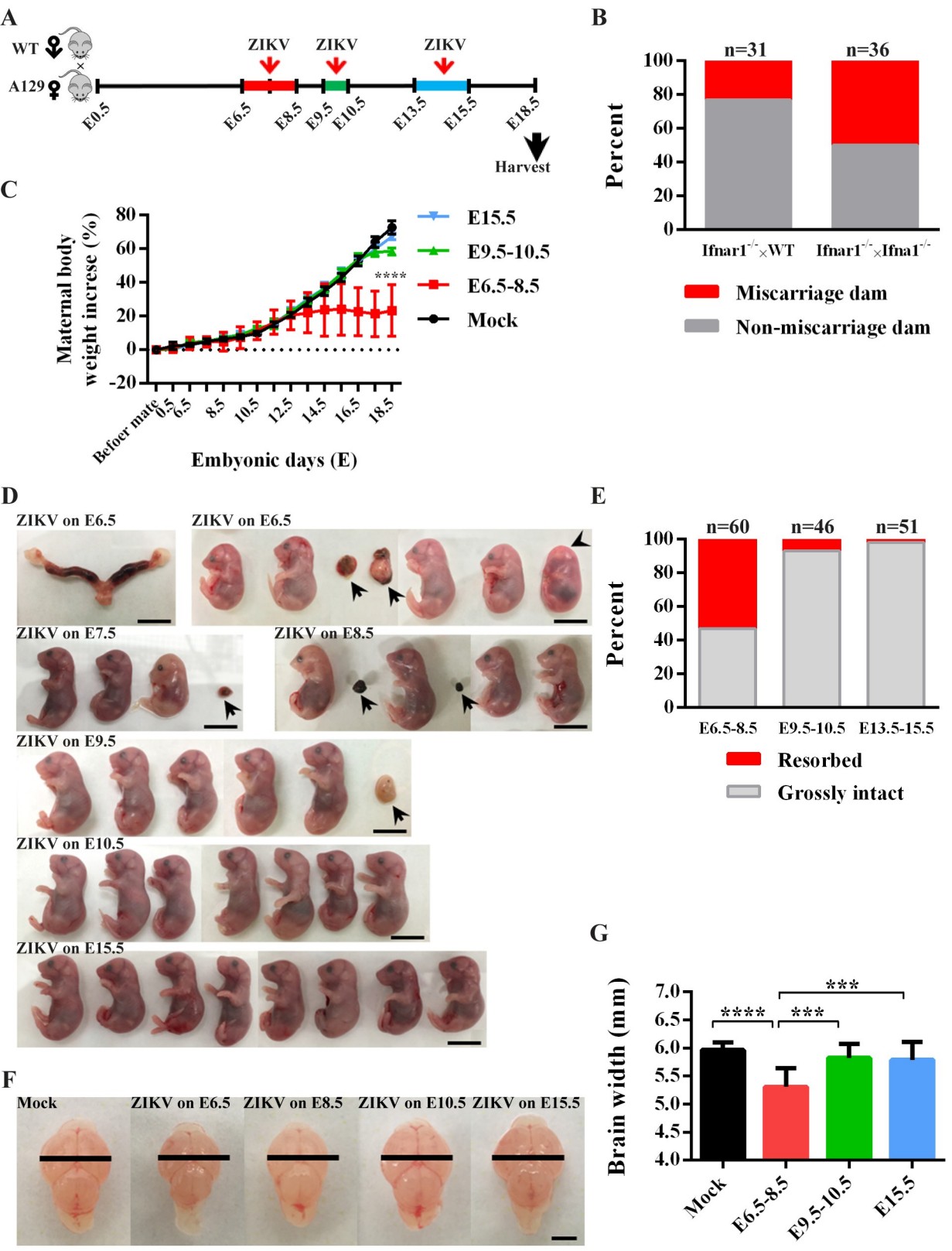

**Fig 1. Gestation stage-dependent ZIKV vertical transmission in a mouse model.** (A) Schematic depiction of ZIKV infection during pregnancy in the mouse model. Ifnar1$^{-/-}$ female mice were crossed with WT males. Pregnant dams were infected with ZIKV at E6.5-E8.5, E9.5-E10.5 or E13.5–15.5. Samples were collected at E18.5. (B) Percentage of miscarriage dams or non-miscarriage dams after maternal infection with ZIKV on E6.5–8.5, E9.5–10.5 or E13.5–15.5. Female Ifnar1$^{-/-}$ mice were crossed to Ifnar1$^{-/-}$ males or WT males. The n above each bar indicates the number of dams per group. Data were assessed by Chi-square test, $P < 0.0001$. (C) Changes in maternal body weight over time in pregnant IFNAR1$^{-/-}$ female mice infected by ZIKV at E6.5-E8.5, E9.5-E10.5 or E15.5. Data represent the mean ± SD. n = 6 in each group. Two-way ANOVA followed by a Tukey post hoc test. (D) Representative uteri and E18.5 fetuses from ZIKV-infected dams. One dam infected by ZIKV on E6.5 underwent miscarriage. Other dams generated grossly intact fetuses, but six of them underwent resorption (arrows in PRV on E6.5 to E9.5). One embryo did not have a well-formed eye structure (arrowhead, PRV on E6.5). Scale bar: 1 cm. (E) Percentage of resorbed or grossly intact fetuses after maternal infection with ZIKV on E6.5–8.5, E9.5–10.5 and E13.5–15.5. The n above each bar indicates the number of pups from 6–8 dams per group. Chi-square test, $P < 0.0001$. (F) Representative images of fetal brains (E18.5) from mock-infected, or dams with ZIKV infection on E6.5, E8.5, E10.5 and E15.5. Large black bars represent the average width of the fetal brain from uninfected dams. Scale bar: 2 mm. (G) Quantification of fetal brain width. Data represent the mean ± SD. n = 15 in mock, n = 21 in E6.5–8.5, n = 20 in E9.5–10.5, n = 23 in E15.5, 3–7 dams per group. Non-parametric Kruskal Wallis test with Dunn's multiple comparisons.

loads were detected in the cortex and serum of dams subjected to early stage infection (Fig 2D and 2E). Collectively, the results suggest that E6.5–8.5 is a critical stage for maternal ZIKV infection that leads to fetal demise and brain malformation in mice.

## Yolk sac-derived microglial progenitors were susceptible to ZIKV infection *in vivo*

Since the infection window E6.5–8.5 is within the critical stage of microglial development and migration, we asked whether murine YS microglia contribute to ZIKV dissemination to the fetal brain. Microglia arise predominantly from erythro-myeloid progenitors in the embryonic YS, which is an extra-embryonic membrane tissue with dense capillary networks (Fig 3A and 3B) and the first site of hematopoiesis in both mice and humans [31, 32]. In mice, the YS-derived microglial progenitors appear at E7 to E8 and migrate to the brain as early as E9.5 after the blood vessels from the YS to the brain are established [33–35]. These microglial cells could be detected in the YS by immunostaining with antibodies targeting the microglia/macrophage lineage markers F4/80 (Fig 3C) and CX3CR1 (Fig 3D). Following ZIKV (the parental PRVABC59 strain, 10$^6$ PFU) infection at E6.5, we found impaired blood vessel formation in the YS, and the number of F4/80$^+$ cells in the YS was significantly reduced (Fig 3B, 3C and 3E). The YS contained F4/80$^+$ microglia co-stained with anti-ZIKV E protein, and many of these ZIKV-infected microglia were located linearly or within the vessel-like structure (Fig 3C and Fig 4A–4C). In the embryonic brain from ZIKV-infected dams (Fig 5A), the number of F4/80$^+$ cells was also decreased, and some F4/80$^+$ microglia were found co-stained with anti-ZIKV E protein (Fig 5B–5D). These data suggest that F4/80$^+$ microglia may be able to carry ZIKV when they invade the fetal brain.

## Yolk sac-derived microglial progenitors were susceptible to ZIKV infection *in vitro*

To confirm that YS-derived microglial progenitors are susceptible to ZIKV infection, we isolated microglial progenitors from the YS on E10.5 (Fig 6A) and infected with ZIKV for one hour. Immunostaining results showed that over 99% of the cells were F4/80$^+$ one week after culture, indicating they were microglia (Fig 6B and 6C). The co-localization of F4/80 with anti-ZIKV antibody demonstrated that cultured microglial progenitors from the YS were susceptible to ZIKV infection (Fig 6B). The infected microglia progenitors were able to produce progeny viruses and release them to the culture medium (Fig 6D). Further quantitative analysis showed that less than 1% of F4/80$^+$ cells were co-stained with anti-ZIKV antibody one week after infection (Fig 6C). These data suggest that F4/80$^+$ microglia may be able to disseminate the viral infection when they invade the fetal brain.

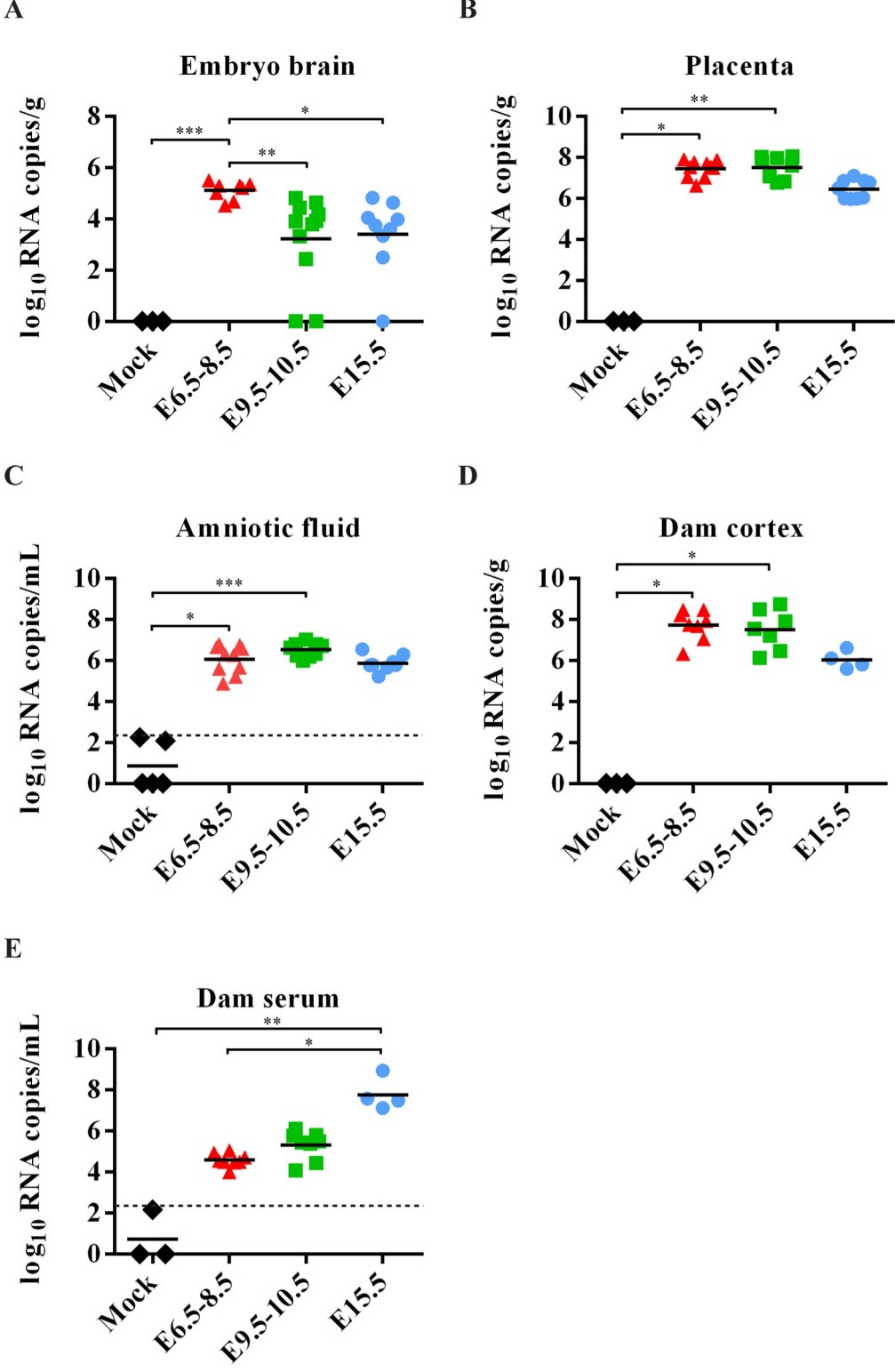

**Fig 2. ZIKV RNA loads in the mouse tissues.** (A-E) ZIKV RNA loads in embryonic brains (n = 3 in mock, n = 8 in E6.5–8.5, n = 11 in E9.5–10.5, n = 9 in E15.5, 3–5 dams per group) (A), placenta (n = 3 in mock, n = 10 in E6.5–8.5, n = 8 in E9.5–10.5, n = 9 in E15.5, 3–4 dams per group) (B), amniotic fluid (n = 5 in mock, n = 10 in E6.5–8.5, n = 9 in E9.5–10.5, n = 8 in E15.5, 3–5 dams per group) (C), dam cortex (n = 3 in mock, n = 8 in E6.5–8.5, n = 7 in E9.5–10.5, n = 4 in E15.5) (D) and dam serum (n = 3 in mock, n = 8 in E6.5–8.5, n = 8 in E9.5–10.5, n = 4 in E15.5) (E), measured

by RT-qPCR. The dashed line represents the limits of detection (LOD). Non-parametric Kruskal Wallis test with Dunn's multiple comparisons. * $P <$0.05, ** $P <$0.01, *** $P <$0.001 and **** $P <$0.0001.

## Ablation of microglial progenitors reduced the viral load in embryonic mouse brains

Microglial development and maintenance require the colony stimulating factor 1 receptor (CSF1R) [36]. To determine whether YS-derived microglial progenitors contribute to ZIKV vertical transmission from mother to fetal brain, anti-CSF1R antibody was injected intraperitoneally at E6.5 and E7.5 into the pregnant mice to ablate the microglial lineage [37] (Fig 7A). As shown in Fig 7B and 7C, two days of anti-CSF1R treatment almost completely depleted microglial progenitors and microglial cells in the YS on E11.5. The ablation efficiency was 99.4% confirmed by staining with the F4/80 marker (Fig 7C), which was consistent with the 99% ablation efficiency previously reported by others [37, 38]. Since the microglia in the fetal brain are solely derived from YS-derived microglia progenitors, as expected, the number of F4/80$^+$ cells in the embryonic brain was also significantly decreased by 90% (Fig 7B and 7D). We then carried out the experiment with pregnant mice receiving the anti-CSF1R antibody at E6.5 and E7.5, accompanied by ZIKV infection within the window of E6.5-E8.5 (Fig 7A). Interestingly, depletion of microglial progenitors reduced ZIKV-caused fetal demise (Fig 7E). ZIKV loads in the YS, embryo brains and hearts were significantly reduced in animals with microglial progenitors depleted (Fig 8A–8C), despite the high viral loads found in the dam's serum and cortex as well as placenta (Fig 8D–8F). These results indicate that YS microglial progenitors may act as "Trojan horses" to transport Zika virus from the mother to the fetal brain.

## Discussion

In this study, we carefully examined the teratogenic effect of ZIKV infection during different stages of pregnancy in a mouse model. The higher risk of ZIKV-mediated abnormal embryonic development during a defined window of pregnancy led us to investigate the role of microglia in the fetal brain dissemination of ZIKV. Our data suggest that YS-derived microglia may serve as a "Trojan horse" to disseminate ZIKV into embryo brains.

In our ZIKV vertical transmission model, we crossed Ifnar1$^{-/-}$ female mice with WT male mice, and found that 77% of the pregnant mice produced Ifnar1$^{+/-}$ fetuses when examined on E18.5 after ZIKV infection. On the other hand, only 50% of Ifnar1$^{-/-}$ female mice, when crossed with Ifnar1$^{-/-}$ males, yielded live Ifnar1$^{-/-}$ fetus on E18.5, indicating a higher miscarriage rate occurred in Ifnar1$^{-/-}$ females when crossed with Ifnar1$^{-/-}$ males. Interestingly, however, the Ifnar1$^{-/-}$ fetuses exhibited better developmental outcomes than the Ifnar1$^{+/-}$ fetuses (from Ifnar1$^{-/-}$×Ifnar1$^{-/-}$: 90.0%, or 108 grossly intact fetuses among total 120 fetuses from 18 dams; from Ifnar1$^{-/-}$×WT: 71.9%, or 128 grossly intact fetuses among total 178 fetuses from 24 dams). This result is similar to the findings of Yockey et al. [39] in terms of fetal survival. Assessing the outcomes of fetuses in the Ifnar1$^{-/-}$×WT model after ZIKV maternal infection, we found that E6.5-E8.5 was a critical stage which led to the increased rate of ZIKV-associated fetal demise and microcephaly. Following infection in later stage, ZIKV was less able to gain access to the fetal brain. These findings are consistent with studies in pregnant women, which have shown that ZIKV infection at early gestational stages increases the risk of miscarriage and fetal brain malformation [1, 2]. Similar results have also been reported in several studies of ZIKV vertical transmission using mouse models [6, 9, 10]. On the other hand, some groups have shown that maternal infection at E9-E14 may also lead to a high rate of abortion,

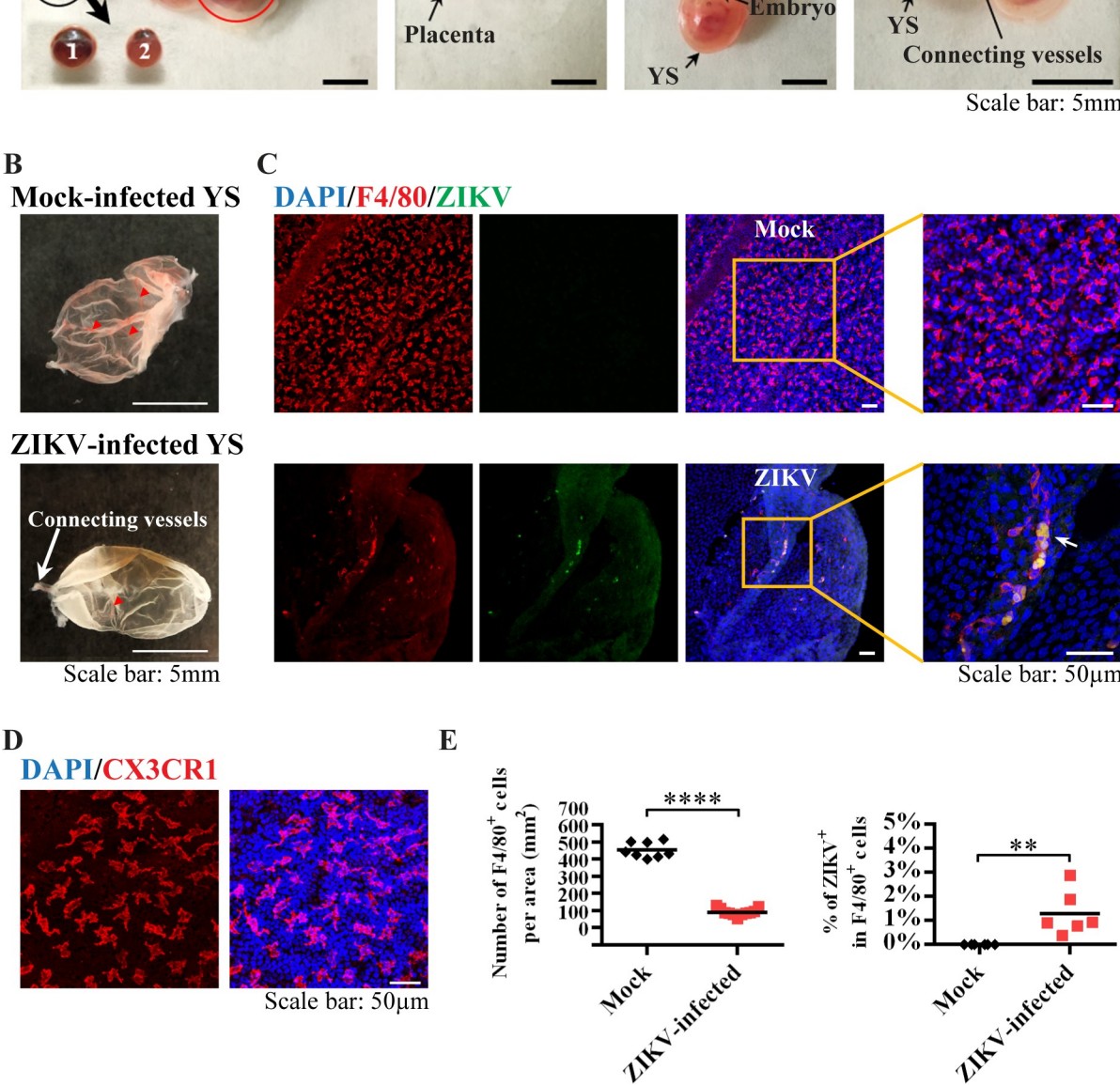

**Fig 3. Characterization of YS-derived microglia progenitors after ZIKV infection *in vivo*.** (A) E11.5 uterus, YS, placenta and embryos from dams infected by ZIKV at E6.5. Two out of five embryos had undergone resorption (embryos 1 and 2), which were taken out from the uterus as the black arrow indicated, shown in (a). One of the other three embryos (embryo 5) was taken from the uterus, as shown in (b). Then, the embryo with the outside YS was carefully separated from the placenta, as shown in (c). In (d), the embryo was separated from the YS. They were connected by vessels. Scale bar: 5 mm. (B) Representative images of E11.5 YS from mock and ZIKV-infected dams on E6.5. Red arrow heads indicate blood vessels in the YS, white arrow indicates connecting vessels. (C) Representative confocal images of YS from mock and ZIKV-infected dam. Microglial progenitors (F4/80, red) co-stained (green) with antibodies against ZIKV E proteins are located linearly (arrow). Scale bar: 50 μm. (D) Representative confocal images of microglial cells (CX3CR1, red) in the E11.5 YS. (E) Quantitative analysis of the numbers of F4/80+ cells per area (n = 8 in mock from 5 dams, n = 11 in ZIKV from 6 dams), and the percentage of ZIKV infected F4/80+ cells in the YS from mock-infected and ZIKV-infected dam (n = 6 in mock from 5 dams, n = 6 in ZIKV from 5 dams). Unpaired t test, ** $P < 0.01$ and **** $P < 0.0001$.

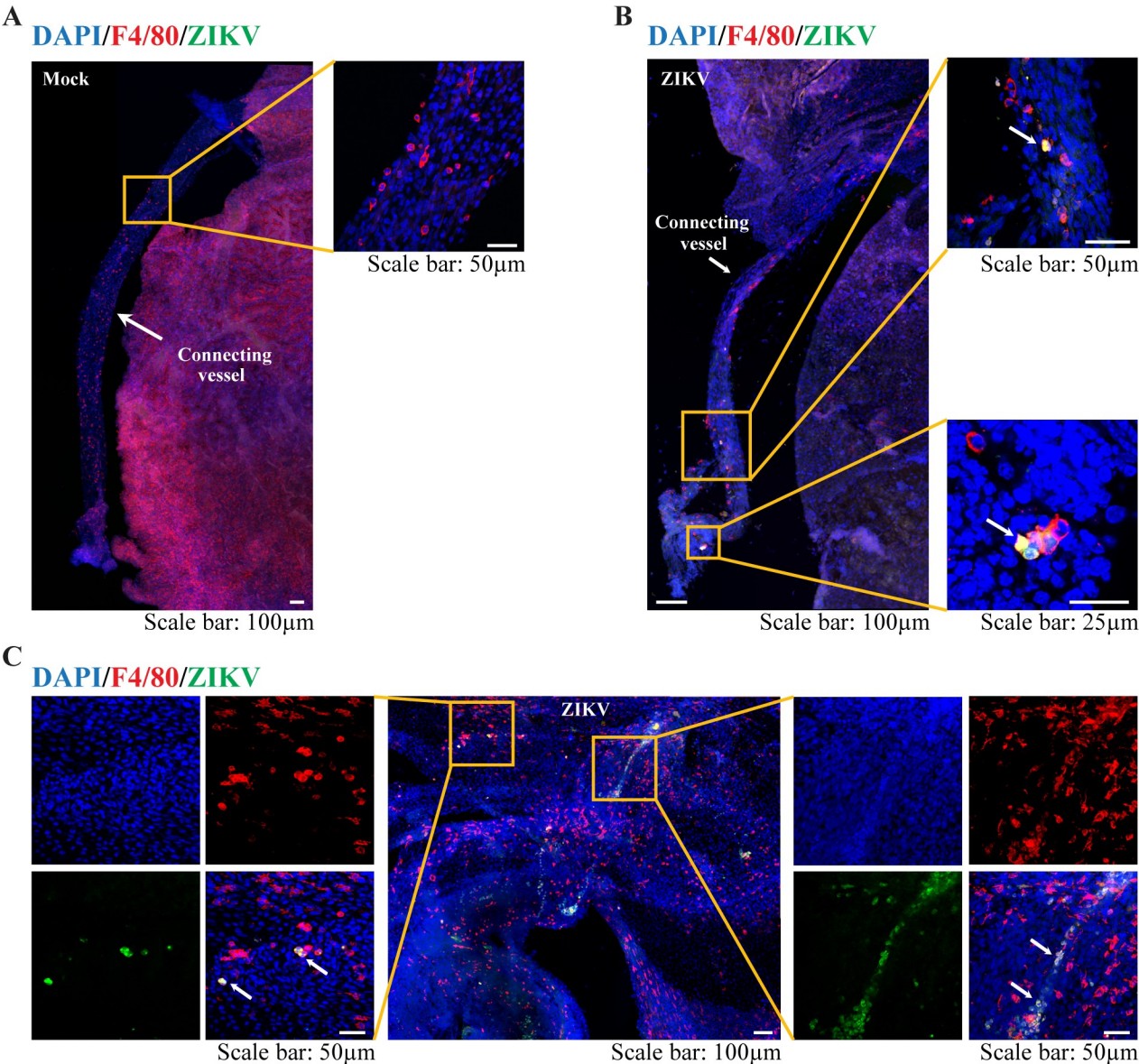

**Fig 4. YS-derived microglia progenitors in the E11.5 YS. (A-C)** Representative confocal images of various regions in the E11.5 YS from mock-infected and ZIKV-infected dam on E6.5. Microglial progenitors (F4/80, red) co-stained with antibodies against ZIKV E proteins (green) (arrow).

significant intrauterine growth restriction and smaller fetal brains [7, 8, 40, 41]. Such a discrepancy may be attributed to different infection routes (intrauterine, intravenous or intraperitoneal), mouse genetic background, viral strains and infection doses. Particularly, intrauterine infection bypasses most of the host defense mechanisms, which likely affects the outcome of fetuses.

The mechanisms of the gestational age-dependent variation in fetal damage following ZIKV vertical transmission have not been well elucidated. The placenta is the primary barrier between the mother and the fetus throughout pregnancy [42]. A fully functional placenta is formed on E10.5 in mice and at the end of the first trimester in humans [43]. ZIKV can traffic across the placenta to reach the fetus by infecting placental cells (villous trophoblasts, fibroblasts, Hofbauer cells and fetal endothelial cells) and compromising the placental barrier [6,

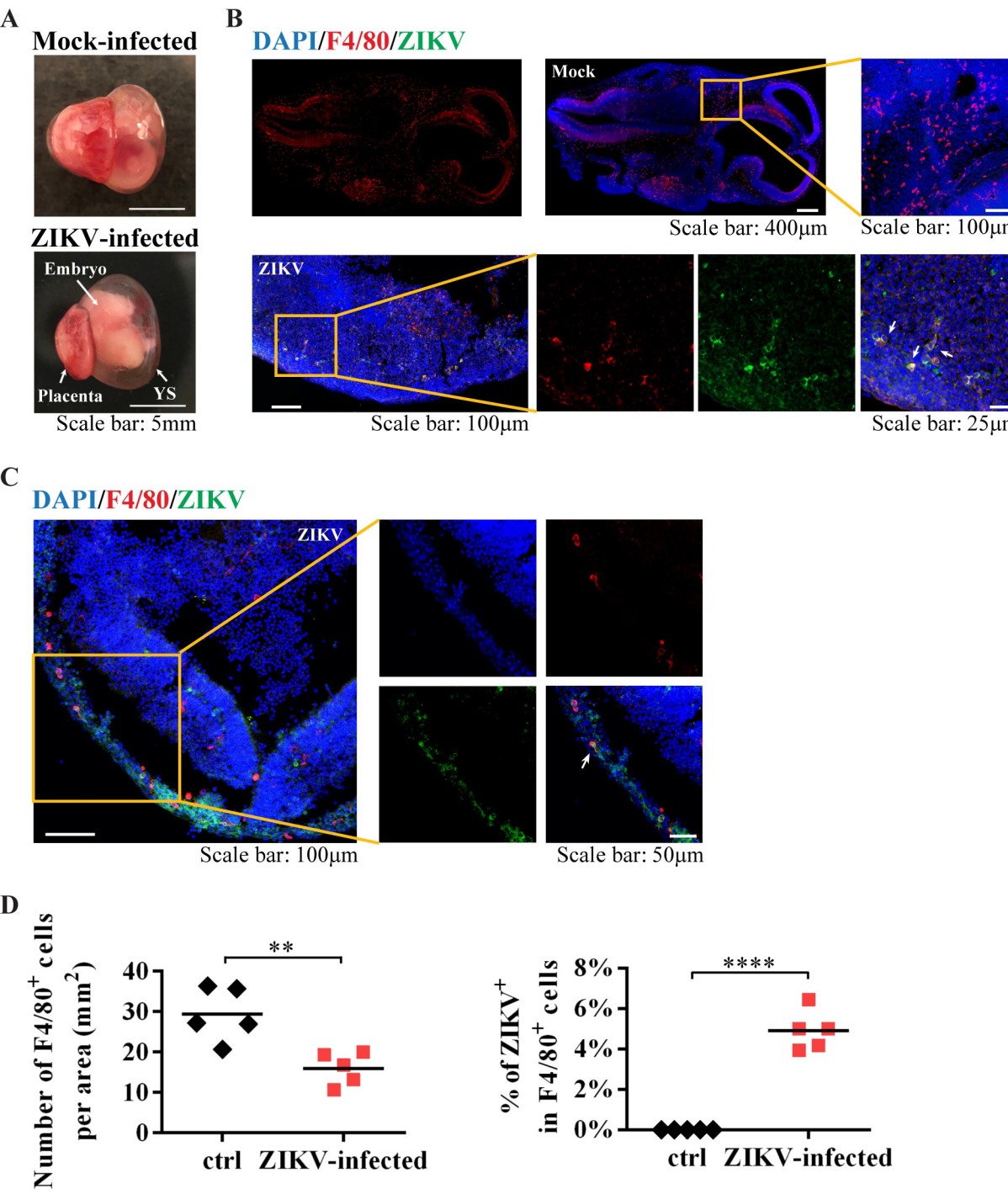

**Fig 5. YS-derived microglia progenitors in the E11.5 embryonic brain.** (A) Representative images of E11.5 embryos with the outside YS and placenta from mock-infected and ZIKV-infected dam on E6.5. (B and C) Representative confocal images of embryonic brains from mock and ZIKV-infected dams. Microglial progenitors (F4/80, red) co-stained with antibodies against ZIKV E proteins (green) (arrow). (D) Quantitative analysis of the numbers of F4/80+ cells per area (n = 5 in mock from 5 dams, n = 5 in ZIKV from 5 dams), and the percentage of ZIKV infected F4/80+ cells in the embryonic brain from mock-infected and ZIKV-infected dam (n = 5 in mock from 5 dams, n = 5 in ZIKV from 3 dams). Unpaired t test, ** $P < 0.01$ and **** $P < 0.0001$.

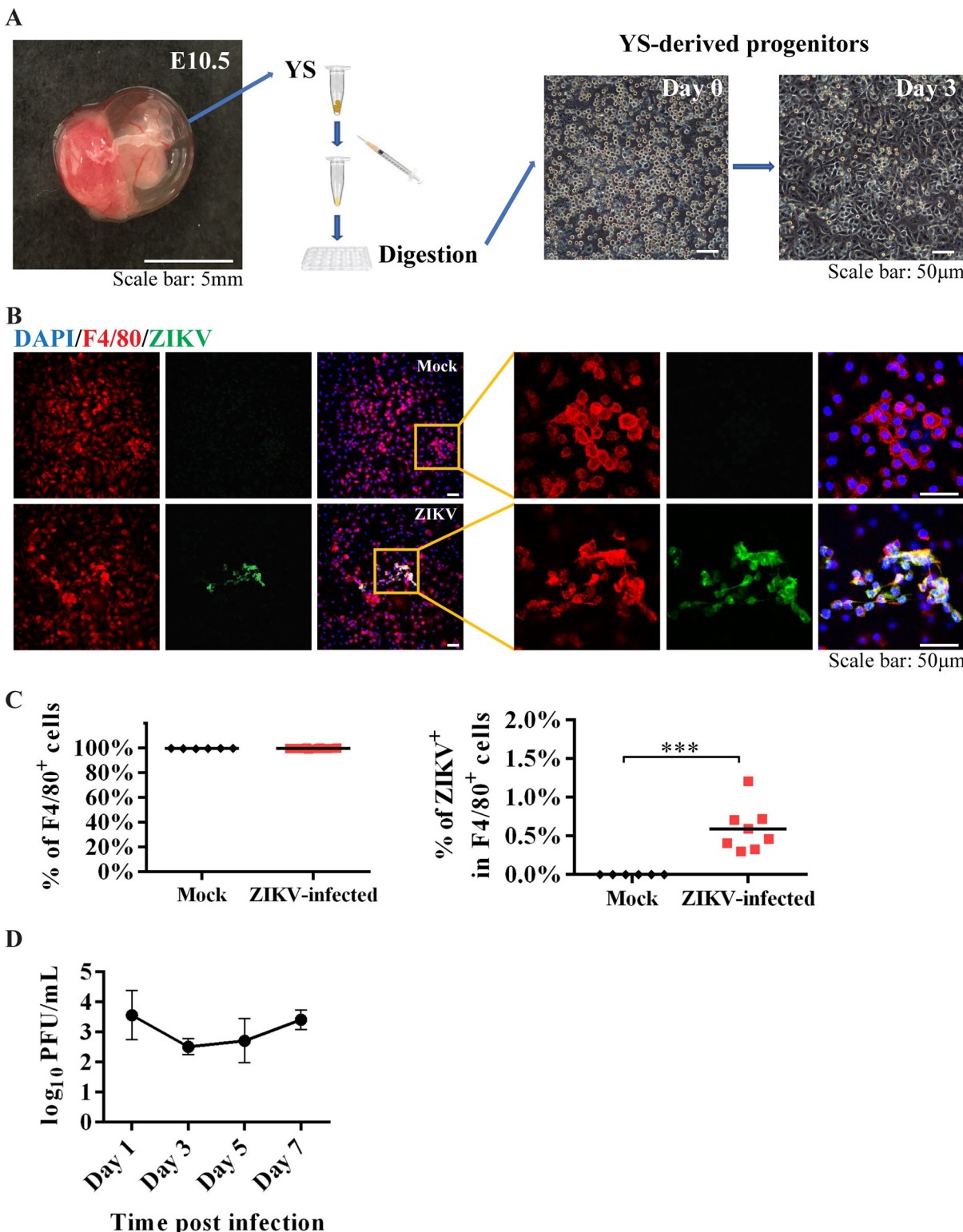

**Fig 6. Characterization of YS-derived microglia progenitors after ZIKV infection *in vitro*.** (A) Isolation of microglial progenitors from YS on E10.5. The YS tissue was isolated into single-cell suspension via mechanical dissociation and chemical digestion (left panels). Most of the cells touched the bottom of the plate three days after culture (right panels). (B) Representative confocal images of cultured YS-derived microglial progenitors expressing F4/80 (red) from control (top row) or those infected by ZIKV (bottom row). ZIKV was detected by antibodies against the E protein (green). Insets are enlarged to the right, showing cells co-labeled with F4/80 marker and ZIKV E proteins

(yellow). (C) Quantitative analysis of the percentage of F4/80$^+$ cells after one week culture (n = 6 in mock from 5 dams, n = 8 in ZIKV from 4 dams), and the percentage of ZIKV-infected F4/80$^+$ cells after one week culture (n = 6 in mock from 5 dams, n = 8 in ZIKV from 4 dams). Unpaired t test, $^{***}$ P <0.001. (D) Plaque assay of the culture medium collected from ZIKV-infected YS-derived microglia progenitors. Data were present as the mean ± SD (n = 4 from 2 dams).

44–49]. Besides the transplacental route of ZIKV vertical transmission, Tabata et al. also proposed a paraplacental route, in which ZIKV spreads from the parietal decidua to amniochorionic membranes [47]. Viral progeny released into the amniotic fluid may infect vulnerable cells in the fetal skin [50, 51]. In our studies, equally high ZIKV RNA loads were detected in both the placenta and amniotic fluid following infection in all gestation stages. However, significantly higher RNA loads in the fetal brain were only found after ZIKV infection at E6.5-E8.5 as compared to the later stages. This infection window precedes placenta formation and is right within the critical period of microglial development, suggesting a role of microglia in ZIKV vertical transmission before placenta formation.

Mouse microglia originate from YS erythro-myeloid progenitors around E7 to E8, and begin to colonize the brain at E9.5 to E10.5 after the circulation system to the brain is established [33–35, 52, 53]. In humans, the size of the YS progressively increases from 5 to 10 weeks of gestation, after which it gradually decreases and disappears after 12$^{th}$ week of gestation (near the end of the first trimester) [54]. YS-derived microglia expand in the central nervous system and maintain themselves via local proliferation throughout life to become the primary innate immune defenders of the brain [16, 32, 53]. Both brain and hiPSCs-derived macrophages/microglia are susceptible to ZIKV infection [20, 55–58]. In our study, we observed for the first time that mouse YS-derived microglia progenitors can be infected by ZIKV. Additionally, ZIKV-infected microglia entered the vessels in the YS. It is thus possible that the infected YS-derived progenitor cells transport ZIKV from mother to embryo, and spread the infection in the brain after migration.

Here, by using the anti-CSF1R antibody to ablate the microglia progenitors that originate from the YS, we found that ZIKV RNA loads significantly decreased in the YS, the embryonic brain and the heart without affecting the viral loads in the dam tissues. Altogether, these results suggest that YS microglial progenitors serve as "Trojan horses", contributing to the ZIKV transfer from the mother to the fetal brain before the complete formation of placenta. An alternative explanation could be that the reduction of ZIKV viral load in the embryo brains after microglia ablation was simply due to the loss of a ZIKV-tropic cell type in the brain. However, this seems unlikely because the degree of viral reduction in embryo brains did not match the extent of microglia depletion. It should be noted that the depletion of YS-derived microglia progenitors did not completely eliminate ZIKV loads in the YS and the embryonic brain. This may be due to that anti-CSF1R-mediated ablation cannot completely remove YS microglia progenitors. Alternatively, other progenitor cells in the YS may also act as transporters during migration if they are susceptible to ZIKV infection [59].

Microglia, besides their immune defensive function, also play an important role in brain development and behavior function [16, 37, 60–62]. Microglia can modulate normal embryonic brain development by affecting the proliferation of NS/PCs, particularly during the peak of NS/PCs proliferation and differentiation [60]. Thus, ablation of microglia could, in theory, reduce the number of NS/PCs and consequently the ZIKV RNA load. However, microglia start to migrate into embryo brains at E9-10, and the effect of microglia ablation on NS/PCs proliferation is minimal by the time we collected the embryos (E11.5) [18]. Thus, the reduced numbers of microglia and possibly NS/PCs are unlikely to count for the degree of ZIKV RNA reduction detected in the E11.5 embryo brains. On the other hand, anti-CSF1R mediates only

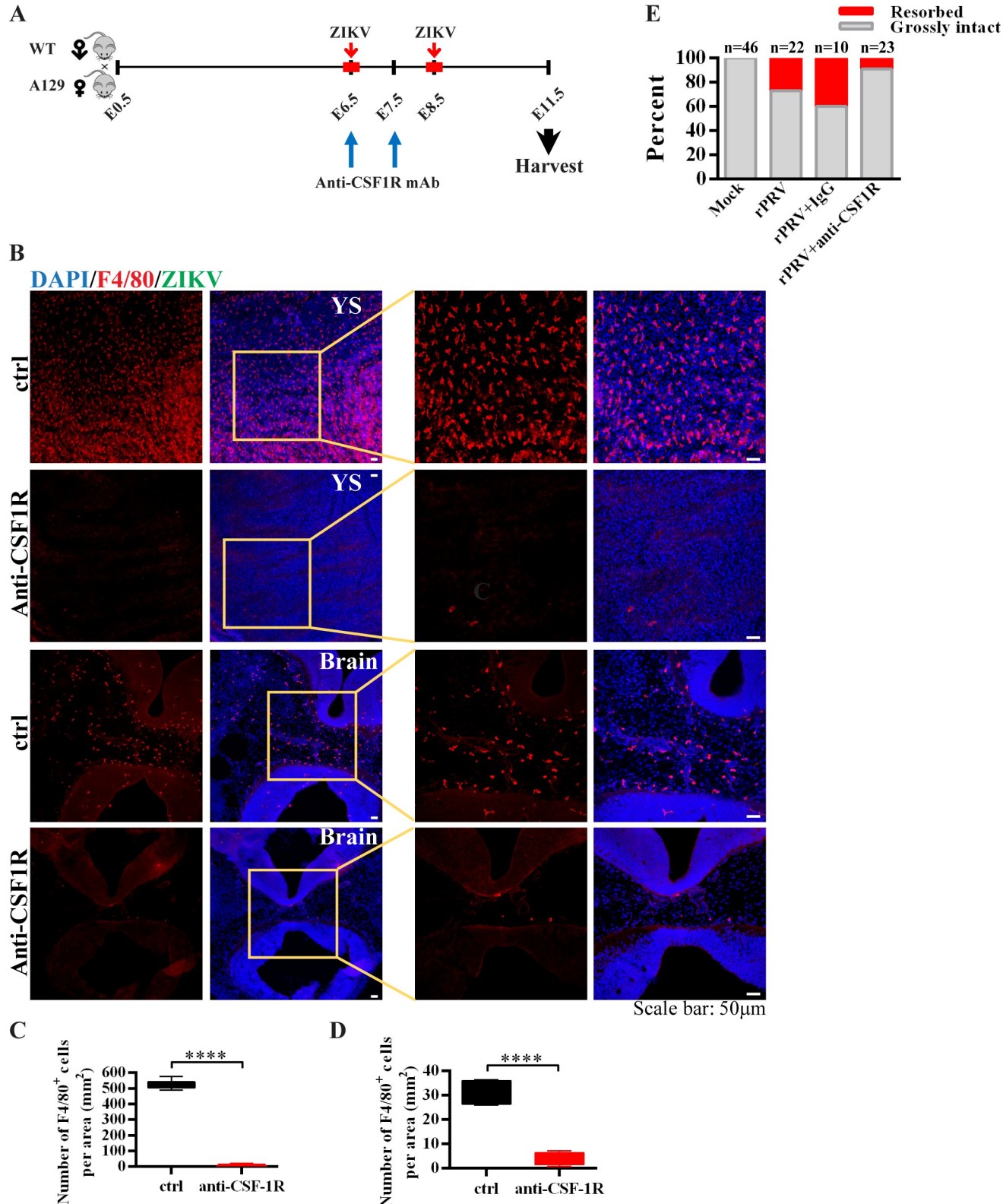

**Fig 7. Depletion of YS-derived microglia progenitors after ZIKV maternal infection.** (A) Illustration of the experimental design. Anti-CSF1R antibody was injected intraperitoneally at E6.5 and E7.5 into pregnant mice accompanied by ZIKV subcutaneous inoculation at E6.5 or E8.5. Samples were collected at E11.5. (B) Representative confocal images of microglial cells (F4/80, red) in the E11.5 YS and the E11.5 embryonic brains with or without anti-CSF1R

treatments. Insets are enlarged to the right. Scale bar: 50 μm. (C) Quantitative analysis of the numbers of F4/80⁺ cells per area in the YS (n = 7 in mock from 4 dams, n = 14 in ZIKV from 4 dams). Unpaired t test, **** *P* <0.0001. (D) Quantitative analysis of the numbers of F4/80⁺ cells per area in the embryonic brain (n = 5 in mock from 5 dams, n = 7 in ZIKV from 3 dams). Unpaired t test, **** *P* <0.0001. (E) Percentage of resorbed or grossly intact fetuses after maternal infection with ZIKV on E6.5–8.5 with or without YS-derived microglia progenitors ablation. The n above each bar indicates the number of pups from 2–7 dams per group. Chi-square test, *P* <0.0001.

a temporary depletion of the YS and brain microglia, which will gradually come back after E14.5 and repopulate completely by the first postnatal weeks [37]. Although not causing significant brain malformation, the microglia ablation during early embryonic stages could induce long-lasting effects on offspring, such as altering the wiring of forebrain circuits and causing hyperactivity or anxiolytic-like behavioral problems [37, 62]. As anti-CSF1R efficiently depletes microglia in both the YS and the embryonic brain, the inflammatory status of microglia per se may have minimal if any effect on brain development in ZIKV-infected embryos. It will be interesting to determine whether anti-CSF1R could directly reduce ZIKV infection/replication in microglia. Easley-Neal et al. [63] reported that these antibodies could cross the blood-brain barrier to exert their function through intraperitoneal injection in adult mice. Our data show a lack of changes of ZIKV RNA in the dam brain 3–5 days after anti-CSF1R antibodies treatment (Fig 4I), indicating that the direct antiviral effect of anti-CSF1R unlikely occurs in this mouse system.

In terms of the effect of microglia depletion on viral infection, Funk et al. demonstrated that CSF1R inhibition-mediated ablation increased WNV infection in adult mice [64]. We found that microglial depletion decreases ZIKV infection in the embryonic brain. This discrepancy is most likely attributed to the different populations of microglia with different functional maturation from early developmental (embryo) vs. fully matured (adult) brains [65]. Before E11.5, the microglia in the fetal brain are solely derived from the YS erythro-myeloid progenitors [18]. In adult mice, monocyte may infiltrate to the brain under inflammation [64, 66, 67]. Monocyte generation begins after E11.5 from the hematopoietic stem cells in the fetal liver and later in the bone marrow [68, 69]. Thus, the E11.5 embryonic brain in our study contains only microglia originated and migrated from the YS; whereas the fully developed adult brain contains both resident microglia and infiltrated monocytes derived from the bone marrow under certain circumstances. The latter, reported by Funk et al. [64], also contributes to the virologic control in the CNS.

Modeling neurotropic virus infection is challenging. To mimic the dynamic interaction between host and virus in human, several animal models have been used. However, the host responses to viral infection may not be fully shared across species, and there may be species-specific differences in viral susceptibility. For example, immunocompetent mice are rather resistant to ZIKV infection because ZIKV NS5 is unable to degrade the antiviral molecule STAT2 in mice [70, 71]. Genetic engineering to remove type I or/and type II IFN signaling yielded several mouse models that are susceptible to ZIKV disease. In our Ifnar1$^{-/-}$×WT mouse model, the miscarriage rate was 23%, which was close to the nonhuman primates model reported by Dudley et al. [30]. The hSTAT2 knock-in immunocompetent mouse model ensues the host immune response, so it is more closely reflects the human disease [72]. However, mice infected with ZIKV do not show Guillain-Barré syndrome as in human [70]. In addition, the structure of the mouse placenta is different from that of humans, which may require higher maternal viremia [70, 73]. Regarding the non-human primate models, several groups reported that ZIKV infection during early pregnancy resulted in fetal death and exhibited many features of congenital Zika syndrome in human [30, 74, 75]. However, non-human primate models are very costly, which limits the number of animals studied. In any case, further studies in an immunocompetent mouse model such as the recently developed hSTAT2

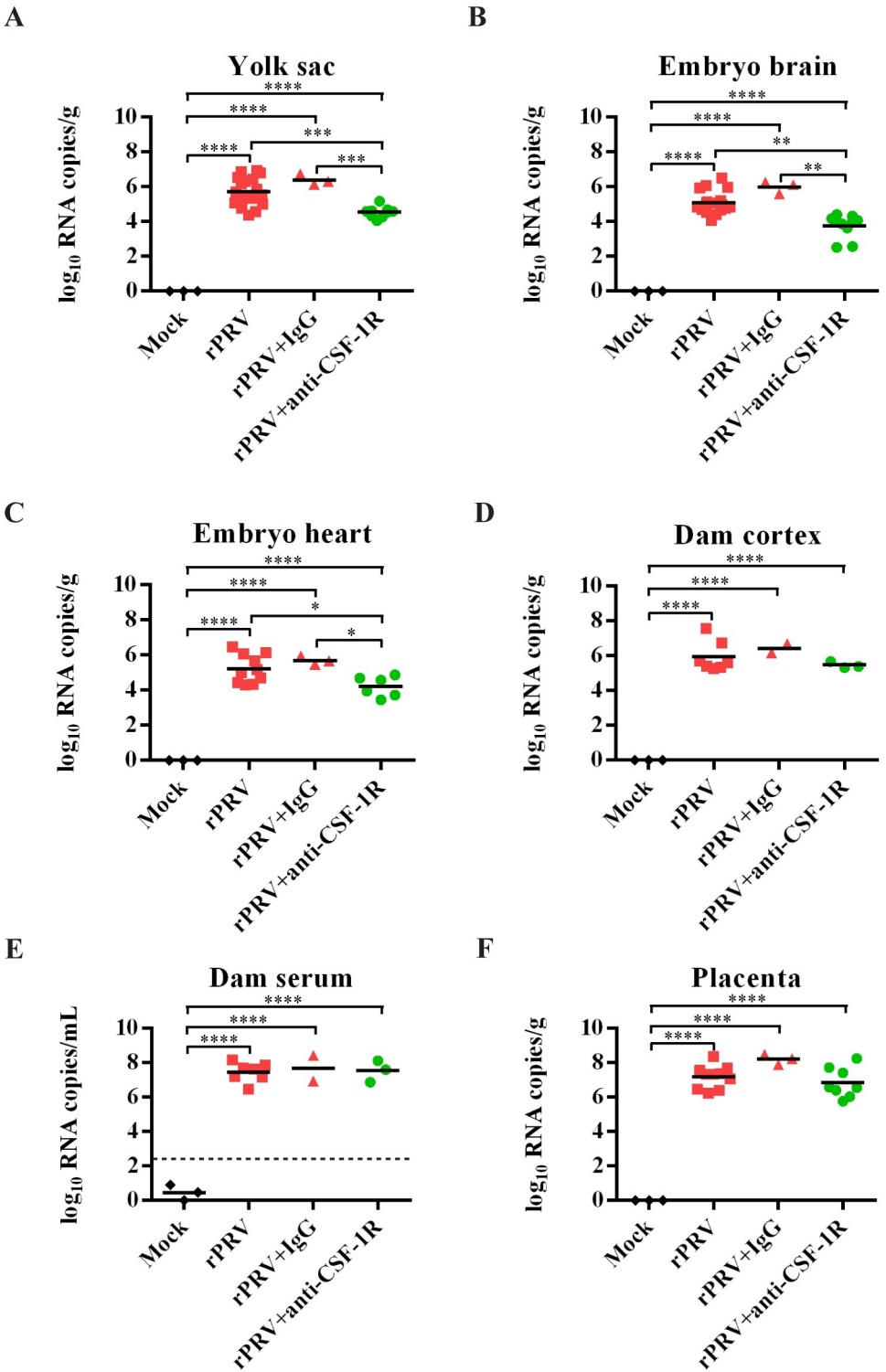

**Fig 8. Viral burden after ZIKV maternal infection with depletion of YS-derived microglia progenitors.** (A-F)
ZIKV RNA loads in the YS (n = 3 in mock, n = 19 in rPRV, n = 3 in rPRV+IgG, n = 9 in rPRV+anti-CSFR1R, 2–7
dams per group) (A), embryonic brain (n = 3 in mock, n = 17 in rPRV, n = 3 in rPRV+IgG, n = 10 in rPRV+anti-
CSFR1R, 2–6 dams per group) (B), embryonic hearts (n = 3 in mock, n = 10 in rPRV, n = 3 in rPRV+IgG, n = 6 in
rPRV+anti-CSFR1R, 2–5 dams per group) (C), dam cortex (n = 3 in mock, n = 7 in rPRV, n = 2 in rPRV+IgG, n = 3 in

rPRV+anti-CSFR1R, 2–7 dams per group) (D), dam serum (n = 3 in mock, n = 7 in rPRV, n = 2 in rPRV+IgG, n = 3 in rPRV+anti-CSFR1R, 2–7 dams per group) (E), and the placenta (n = 3 in mock, n = 10 in rPRV, n = 3 in rPRV+IgG, n = 8 in rPRV+anti-CSFR1R, 2–3 dams per group) (F), measured by RT-qPCR. One–way ANOVA with a Tukey post hoc test. The dashed line represents the LOD. * $P <$0.05, ** $P <$0.01, *** $P <$0.001 and **** $P <$0.0001.

knock-in mice or in the non-human primate model are warranted to confirm the role of yolk sac microglia as a general mechanism mediating ZIKV dissemination into embryonic brains.

## Author Contributions

**Conceptualization:** Pei Xu, Xuping Xie, Shannan L. Rossi, Nikolaos Vasilakis, Pei-Yong Shi, Scott C. Weaver, Ping Wu.

**Formal analysis:** Pei Xu, Hongjie Xia, Ping Wu.

**Funding acquisition:** Pei Xu, Javier Allende Labastida, Nikolaos Vasilakis, Pei-Yong Shi, Scott C. Weaver, Ping Wu.

**Investigation:** Pei Xu, Chao Shan, Tiffany J. Dunn, Xuping Xie, Hongjie Xia, Junling Gao, Javier Allende Labastida, Jing Zou, Paula P. Villarreal, Caitlin R. Schlagal, Yongjia Yu, Gracie Vargas, Shannan L. Rossi.

**Methodology:** Pei Xu, Chao Shan, Tiffany J. Dunn, Hongjie Xia, Gracie Vargas, Shannan L. Rossi, Ping Wu.

**Resources:** Shannan L. Rossi, Nikolaos Vasilakis, Pei-Yong Shi, Scott C. Weaver, Ping Wu.

**Supervision:** Gracie Vargas, Shannan L. Rossi, Nikolaos Vasilakis, Pei-Yong Shi, Scott C. Weaver.

**Validation:** Pei Xu.

**Visualization:** Pei Xu, Jing Zou, Ping Wu.

**Writing – original draft:** Pei Xu.

**Writing – review & editing:** Pei Xu, Chao Shan, Tiffany J. Dunn, Xuping Xie, Junling Gao, Javier Allende Labastida, Jing Zou, Paula P. Villarreal, Caitlin R. Schlagal, Yongjia Yu, Gracie Vargas, Shannan L. Rossi, Nikolaos Vasilakis, Pei-Yong Shi, Scott C. Weaver, Ping Wu.

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
