## [Decision Letter · Decision Letter 0]

18 Feb 2020

Dear Dr. Wu,

Thank you very much for submitting your manuscript "Role of yolk sac microglia in vertical transmission of Zika virus 

from mother to fetal brain" for consideration at PLOS Neglected Tropical Diseases. As with all papers reviewed by the journal, your manuscript was reviewed by members of the editorial board and by several independent reviewers. In light of the reviews (below this email), we would like to invite the resubmission of a significantly-revised version that takes into account the reviewers' comments. 

In general the reviewers of this manuscript liked this manuscript but have identified several important concerns that need to be addressed prior to further consideration of this submission. I ask that you also give consideration to comments from the reviewers regarding data interpretation and address some of the potential caveats that have been highlighted.

Further, we would like to apologize for the long review process for this submission. It took some time to identify reviewers for this manuscript.

We cannot make any decision about publication until we have seen the revised manuscript and your response to the reviewers' comments. Your revised manuscript is also likely to be sent to reviewers for further evaluation.

Sincerely,

Michael R Holbrook, PhD

Guest Editor

Rebecca Rico-Hesse

Deputy Editor

In general the reviewers of this manuscript liked this manuscript but have identified several important concerns that need to be addressed prior to further consideration of this submission. I ask that you also give consideration to comments from the reviewers regarding data interpretation and address some of the potential caveats that have been highlighted.

Further, I would like to apologize for the long review process for this submission. It took some time to identify reviewers for this manuscript.

Reviewer's Responses to Questions

**Key Review Criteria Required for Acceptance?**

**Methods**

-Are the objectives of the study clearly articulated with a clear testable hypothesis stated?

-Is the study design appropriate to address the stated objectives?

-Is the population clearly described and appropriate for the hypothesis being tested?

-Is the sample size sufficient to ensure adequate power to address the hypothesis being tested?

-Were correct statistical analysis used to support conclusions?

-Are there concerns about ethical or regulatory requirements being met?

Reviewer #1: We have no concerns about the methods used in this study. We would like to encourage the authors to measure infectious virus and not just viral RNA as this does not confirm whether use of the monoclonal antibody actually prevents virus replication in tissues.

Reviewer #2: -The data in figure 1B are very interesting, though the experiment is under-powered with only an n of 4. This experiment should be repeated. To what do the authors attribute the differences in maternal weight gain following early term infection vs late term infection? Importantly, does the virologic status of the dam change depending on the gestational timepoint of infection? These data would clarify the authors’ overall findings considerably and bolster this study’s value as a reference for the field. 

-While the images in Figure 2 are generally of high quality, the reliance on representative images with no quantification diminishes enthusiasm for the study. Quantification of the numbers of infected F4/80+ cells detected per unit area in yolk sac and fetal brain should be provided to establish the level of variability and overall generalizable nature of these findings. 

-In Figure 2C, how were the imaged structures determined to be blood vessels? Why do F4/80+ cells appear to permeate the yolk sack in 2B, but seem concentrated in a single blood vessel in 2C. The differences in cellularity between these two images is confusing. Again, more robust quantification and explanation of analytical methods is necessary to interpret these data. 

-A more quantitative assessment of viral infection in YS microglial progenitors is warranted in Figure 2E. Are these cells productively infected, producing progeny virus? What is the time course of viral burden and shedding? Multistep growth curve or similar analyses would be helpful to more robustly profile the virologic profile of these cells. 

-The efficiency of depletion of microglia in YS and fetal brain following antibody administration to the pregnant dam is somewhat surprising. The authors should include quantification of F4/80+ cells per unit area across multiple mice in figures 3B and 3C to establish the consistency and reproducibility of this depletion. 

-Were microglia also depleted in the brains of dams treated with anti-CSFR1 antibodies? While in theory, perhaps, these antibodies would not cross the BBB, this should be experimentally confirmed. If changes to maternal microglial numbers are discovered, these should be accounted for in the interpretation of the authors’ results.

Reviewer #3: Please include more information on the sample size for each experiment.

Additional information on how representative the images shown are and if there is any way to quantify the level of ablation among the different animals it would strengthen they results.

**Results**

-Does the analysis presented match the analysis plan?

-Are the results clearly and completely presented?

-Are the figures (Tables, Images) of sufficient quality for clarity?

Reviewer #1: Result are clearly presented with exceptionally good images.

Reviewer #2: - The rationale for reporting miscarriage rates in Ifnar1xWT vs. Ifnar1xIfnar 1 crosses (Figure S1) is unclear. Are the authors justifying why they chose to use WT sires for their main studies? At present, these data do not seem relevant and need further discussion. Moreover, these data are contradictory to those reported by Yockey et al (PMID 29305462) who reported worse outcomes in Ifnar1-heterozygous embryos compared to Ifnar1-deficient embryos following ZIKV infection. The authors should discuss this discrepancy.

-Similar to above – what is the purpose of Figure S2? Are the authors justifying using an infectious clone in their study? These data should be more well integrated into the paper or removed. 

-The statistical comparisons in Figures 1G-I appear inconsistent. Some comparisons with smaller effect sizes and higher variance have higher p values than other comparisons with larger effect sizes and less variance. Which test was used for these graphs, specifically? For Figure 1I, why are values reported below the limit of detection?

Reviewer #3: Fig 2B – Add a scale bare to the zoomed images.

Fig 2 – Please add the number of embryo brains processed that showed similar data.

Fig 3 – How consistent was the ablation of YS microglia? Only a single micrograph is shown, is this representative of X embryos, or did some embryos retain higher levels of staining? Is there any way to quantify (RT-PCR in embryo brain?)?

**Conclusions**

-Are the conclusions supported by the data presented?

-Are the limitations of analysis clearly described?

-Do the authors discuss how these data can be helpful to advance our understanding of the topic under study?

-Is public health relevance addressed?

Reviewer #1: The fundamental missing piece of data that results in an incomplete story is whether treatment of dams with anti-CSFIR reverses the detrimental effects of ZIKV on offspring viability (as shown in Figure 1). It is never reported whether by reducing ZIKV in the brain microglia reverses the adverse effects of ZIKV on development or if that is caused by something else (e.g., immunopathology). Along those lines, the authors also do not show the impact of anti-CSFIR alone (without ZIKV) on microglia and adverse outcomes. Finally, there are no data pertaining to microglia activation, only co staining for ZIKV. Does anti-CSFIR reduce not only microglia infection but activation (i.e., neuroinflammation)?

Reviewer #2: -Funk and colleagues recently reported decreased virologic control of WNV-infected mice treated with a CSFR1 antagonist (PMID 30704498). In general, more detailed discussion of previous studies demonstrating the anti-viral capacity of microglia in adults using similar depletion strategies should be discussed, as they contrast somewhat with the authors findings in fetuses in the current study. 

-A major issue of concern is the use of immunodeficient Ifnar-/- animals as the basis for in vivo phenotypes. IFN-deficient mice exhibit greatly increased viremia and expanded viral tropism that may confound the ability to truly map routes of entry into the fetal CNS as they occur naturally. This issue is one faced by the entire field and not solely a problem with the current manuscript, but some circumspection in the interpretations of the authors’ findings is warranted in the discussion. While likely beyond the scope of the current manuscript, establishing similar phenotypes in an immunocompetent mouse model (such as the recently developed hSTAT2 knockin mouse) or even in NHP models will be needed to confirm the authors’ findings. The authors should elaborate on these points in the discussion. 

-While an intriguing hypothesis, the authors do not currently have sufficient evidence to establish any role for microglial progenitors in “vertical transmission.” Any virus present in the fetal YS would in theory already be “transmitted” from the infected dam. Instead, the authors posit a possible route of viral dissemination to the fetal brain, but even this possibility will need more careful experimentation to establish. The authors findings could simply arise from a depletion of a tropic cell type from the fetal brain, thereby diminishing a major replicative niche for the virus, irrespective of the movement of infected cells from YS to fetal brain. The authors should be more circumspect in their interpretation of the findings in figure 3, or otherwise perform more careful time course experiments in which YS microglia are depleted either before or after populating the fetal brain.

Reviewer #3: I assume ablation of microglial progenitor cells during this early time of development has severe consequences to normal embryo brain development. Please discuss how this alteration in brain development could alter ZIKV replication in the brain. What would happen if you ablated the cells and directly inoculated the brain? Would there be less permissive cells there? To further establish these cells are a “Trojan horse” for the virus to get across the maternal-fetal interface, can you quantify virus levels in non-brain tissue of the embryos? Is it similarly decreased or is it specific for virus levels in the brain?

**Editorial and Data Presentation Modifications?**

Reviewer #1: n/a

Reviewer #2: -For the n values for embryos throughout the manuscript, the number of unique litters should also be reported.

Reviewer #3: Fig 1B – Figure is labeled “WT”, but I think it is showing “mock” infected data?

Fig 2A – The Figure legend description of the pictures is somewhat hard to follow. There are several arrows in the picture and two things circled. It may help to number each embryo. Alternatively, if the picture is shown to demonstrate the distinct location of the yolk sac versus the embryo you may want to just include a cartoon and last picture in the series.

**Summary and General Comments**

Reviewer #1: Overall, i like the direction of the studies and the novelty of yolk sac derived microglia. How this impacts adverse outcomes in the fetus as well as microglia activation are logical, yet missing experiments in the current manuscript.

Reviewer #2: In this study, Xu et al present intriguing evidence for the importance of gestational age on the dissemination of ZIKV to the fetal brain, with some exploration of possible roles for yolk sac derived microglia. The authors’ systematic characterization of the virologic outcomes of maternal ZIKV infection at different stages of gestation is highly informative and an important addition to the existing literature. While the authors convincingly demonstrate that fetal microglial progenitors are tropic for ZIKV and contribute to viral burden in the fetal brain, the purported roles for these cells in transmission/dissemination are somewhat prematurely drawn. While this study will likely be of broad interest, some improvements are warranted before publication.

Reviewer #3: This study by Xu et al is characterizing the role of yolk sac microglial progenitor cells in transporting Zika virus across the maternal placenta to infect the developing embryo. Ablation of the yolk sac microglial progenitor cells reduced ZIKV in the embryonic brain, although did not eliminate it. The paper is well written and straight forward. To enhance the conclusions of the study additional discussion about how ablation of the YS microglial progenitor cells will alter the brain structure is warranted.

PLOS authors have the option to publish the peer review history of their article (what does this mean?). If published, this will include your full peer review and any attached files.

Reviewer #1: No

Reviewer #2: No

Reviewer #3: No
---

## [Decision Letter · Decision Letter 1]

12 May 2020

Dear Dr. Wu,

Thank you very much for submitting your manuscript "Role of microglia in the dissemination of Zika virus 

from mother to fetal brain" for consideration at PLOS Neglected Tropical Diseases. As with all papers reviewed by the journal, your manuscript was reviewed by members of the editorial board and by several independent reviewers. The reviewers appreciated the attention to an important topic. Based on the reviews, we are likely to accept this manuscript for publication, providing that you modify the manuscript according to the review recommendations. 

Please move supplemental figures to the main text. PLoS NTDs does not limit the number of figures so it is best to include as much data as is reasonable in the primary publication. Some of the data provided in supplemental material is important for interpretation of you data.

Sincerely,

Michael R Holbrook, PhD

Associate Editor

Rebecca Rico-Hesse

Deputy Editor

Please move supplemental figures to the main text. PLoS NTDs does not limit the number of figures so it is best to include as much data as is reasonable in the primary publication. Some of the data provided in supplemental material is important for interpretation of you data.

Reviewer's Responses to Questions

**Key Review Criteria Required for Acceptance?**

**Methods**

-Are the objectives of the study clearly articulated with a clear testable hypothesis stated?

-Is the study design appropriate to address the stated objectives?

-Is the population clearly described and appropriate for the hypothesis being tested?

-Is the sample size sufficient to ensure adequate power to address the hypothesis being tested?

-Were correct statistical analysis used to support conclusions?

-Are there concerns about ethical or regulatory requirements being met?

Reviewer #1: • The authors have cultured Yolk sac microglia-progenitors, infected them with ZIKV, and then performed a plaque assay. We had intended that the authors would have use a plaque assay or similar measure of infectious virus to evaluate the infectious virus in collected tissue, for example in Figure 1G-I, and Figure 4F-J. In figure 4 specifically, this would allow the authors to show that the Anti-CSFR1 antibody decreased the amount of infectious zika virus in the embryonic brain. The plaque assay experiment in Figure S3 may fit better incorporated into figure 3.

Reviewer #2: (No Response)

Reviewer #3: They have added additional information on sample size, and in some cases increased samples size. No concerns.

**Results**

-Does the analysis presented match the analysis plan?

-Are the results clearly and completely presented?

-Are the figures (Tables, Images) of sufficient quality for clarity?

Reviewer #1: • The authors’ additional experiment (Figure S4A) have addressed our main concern here, and have shown that depletion of the yolk sac progenitors with the anti-CSF1R antibody reduces the fetal demise during zika infection. This figure seems vital to the overall narrative of the manuscript, and we would suggest adding it to figure 4.

Reviewer #2: (No Response)

Reviewer #3: The new version of the figures is easier to understand.

The current figures were all blurry after download - but probably just a conversion problem.

**Conclusions**

-Are the conclusions supported by the data presented?

-Are the limitations of analysis clearly described?

-Do the authors discuss how these data can be helpful to advance our understanding of the topic under study?

-Is public health relevance addressed?

Reviewer #1: fine

Reviewer #2: (No Response)

Reviewer #3: The conclusions are well defined and address the data as well the the public health relevance.

**Editorial and Data Presentation Modifications?**

Reviewer #1: (No Response)

Reviewer #2: (No Response)

Reviewer #3: Accept

**Summary and General Comments**

Reviewer #1: Move data from supplemental information to main figures in text. Also, measuring infectious virus in tissues is needed.

Reviewer #2: (No Response)

Reviewer #3: All concerns were addressed.

PLOS authors have the option to publish the peer review history of their article (what does this mean?). If published, this will include your full peer review and any attached files.

Reviewer #1: No

Reviewer #2: No

Reviewer #3: No
---

## [Editor Report · Decision Letter 2]

22 May 2020

Dear Dr. Wu,

We are pleased to inform you that your manuscript 'Role of microglia in the dissemination of Zika virus

from mother to fetal brain' has been provisionally accepted for publication in PLOS Neglected Tropical Diseases.

Best regards,

Michael R Holbrook, PhD

Associate Editor

Rebecca Rico-Hesse

Deputy Editor

---

## [Editor Report · Acceptance letter]

26 Jun 2020

Dear Dr. Wu,

We are delighted to inform you that your manuscript, "Role of microglia in the dissemination of Zika virus from mother to fetal brain ," has been formally accepted for publication in PLOS Neglected Tropical Diseases.

Best regards,

Shaden Kamhawi

co-Editor-in-Chief

Paul Brindley

co-Editor-in-Chief
